# The Role of E3 Ubiquitin Ligase Gene *FBK* in Ubiquitination Modification of Protein and Its Potential Function in Plant Growth, Development, Secondary Metabolism, and Stress Response

**DOI:** 10.3390/ijms26020821

**Published:** 2025-01-19

**Authors:** Yuting Wu, Yankang Zhang, Wanlin Ni, Qinghuang Li, Min Zhou, Zhou Li

**Affiliations:** College of Grassland Science and Technology, Sichuan Agricultural University, Chengdu 611130, China; wuyuting@stu.sicau.edu.cn (Y.W.); 2023243020@stu.sicau.edu.cn (Y.Z.); niwanlin@stu.sicau.edu.cn (W.N.); 2023343030@stu.sicau.edu.cn (Q.L.); 2021202098@stu.sicau.edu.cn (M.Z.)

**Keywords:** ubiquitination, UPS, E3 ligase, FBK, growth and development, biotic stress, abiotic stress

## Abstract

As a crucial post-translational modification (PTM), protein ubiquitination mediates the breakdown of particular proteins, which plays a pivotal role in a large number of biological processes including plant growth, development, and stress response. The ubiquitin-proteasome system (UPS) consists of ubiquitin (Ub), ubiquitinase, deubiquitinating enzyme (DUB), and 26S proteasome mediates more than 80% of protein degradation for protein turnover in plants. For the ubiquitinases, including ubiquitin-activating enzyme (E1), ubiquitin-conjugating enzyme (E2), and ubiquitin ligase (E3), the FBK (F-box Kelch repeat protein) is an essential component of multi-subunit E3 ligase SCF (Skp1-Cullin 1-F-box) involved in the specific recognition of target proteins in the UPS. Many *FBK* genes have been identified in different plant species, which regulates plant growth and development through affecting endogenous phytohormones as well as plant tolerance to various biotic and abiotic stresses associated with changes in secondary metabolites such as phenylpropanoid, phenolic acid, flavonoid, lignin, wax, etc. The review summarizes the significance of the ubiquitination modification of protein, the role of UPS in protein degradation, and the possible function of *FBK* genes involved in plant growth, development, secondary metabolism, and stress response, which provides a systematic and comprehensive understanding of the mechanism of ubiquitination and potential function of *FBKs* in plant species.

## 1. Introduction

Protein turnover and post-translational modifications (PTMs) regulate the amount of proteins and their activities during plant growth and development [1,2]. The structure and function of plant proteins can be altered by the PTMs, including ubiquitination, phosphorylation, acetylation, glycosylation, and SUMOylation, etc., as a result of changes in amino acid charge, spatial effect, and the activity of key catalytic residue [2,3,4,5,6,7]. These diverse PTMs impact the protein spatial structure, stability, subcellular localization, and interaction with other proteins, thereby regulating multiple pathways such as DNA damage repair, gene expression, protein stability, and phytohormone signaling [3,4,5]. Protein PTM has been found to be one of the fastest responses to changing environmental conditions through activating cellular signal transduction pathways, because this process is not involved in the de novo synthesis of proteins [2,8]. Moreover, physiological changes cause interactions among different types of PTMs to form a complex crosstalk network, thereby mediating plant growth, development, and stress tolerance [9,10,11,12]. Compared with more than 650 PTMs in animals [13], plants have a smaller variety of PTMs, of which only 33 different PTMs are identified up to now [14]. As one of the most common PTMs in plants, ubiquitination plays an important role in cellular protein metabolism, plant growth and development, the biosynthesis of secondary metabolite, and adaptive response to various environmental stresses [15,16]. The ubiquitin–26S proteasome system (UPS) consists of ubiquitin (Ub), ubiquitinase, deubiquitinating enzyme (DUB), and 26S proteasome, which regulates the degradation of more than 80% proteins by binding Ub to target proteins in plant species [1,17]. These target proteins are firstly tagged by Ub molecules via the catalytic reaction of ubiquitinases and then recognized and broken down by the 26S proteasome [18].

The ubiquitin-activating enzyme (E1), ubiquitin-conjugating enzyme (E2), and ubiquitin-ligating enzyme (E3) are three critical components of ubiquitinase [19]. Genes encoding E1 and E2 enzymes are relatively conservative in plant genomes [11,20]. In contrast, E3 enzymes are encoded by a large diversity of genes. For example, more than 1500 E3 enzymes and only 6 E1 enzymes and 49 E2 enzymes exist in rice (*Oryza sativa*) [7,21]. The E3 ubiquitin ligases are primarily responsible for the specific recognition of target proteins in the UPS and can be categorized into single-subunit and multisubunit proteins based on their structural characteristics [1,22]. As a key component of the multisubunit E3 ligase SCF (Skp1-Cullin 1-F-box) complex, FBK (F-box Kelch repeat protein) is rich in Kelch structure which is responsible for the recognition of target proteins. The *FBK* gene family has been increasingly identified and reported in plants in recent years associated with the regulation of plant growth, developments of flower, leaf, and seed, secondary metabolism, and adaptive response to adverse environments [1,23,24]. The review summarizes the significance of ubiquitination and the possible function of *FBK* genes in plant growth, secondary metabolism, and stress response.

## 2. Ubiquitination Modification of Protein in Plants

As one of the major PTMs in plants, the role and function of ubiquitination has become a hotspot in biological research nowadays [12]. There are two types of ubiquitination processes including mono- and poly-ubiquitination (Figure 1). For mono-ubiquitination, an Ub is added to the residue of target protein to modify protein function (Figure 1a). Multiple Ub monomers can attach to several different lysines in a protein at the same time, which is called multimono-ubiquitination (Figure 1b). Polyubiquitination means that multiple Ubs are linked to their own lysine residues to form Ub chains on the amino acid residue of a target protein (Figure 1c,d) [11,22,25,26]. Linear polyubiquitination and branching polyubiquitination are two types of polyubiquitination. When the Ub chain linked to a substrate protein is just a single chain without any extra branches, the process is defined as linear polyubiquitination (Figure 1c). Branching polyubiquitination indicates that Ubs on the Ub chain also connect with other Ubs, thus creating branching chains (Figure 1d) [22,26]. These Ub-tagged proteins are then specifically recognized and degraded by different proteasomes [22,27,28,29,30]. Ubiquitination widely participates in DNA damage repair, cell cycle, protein abundance and activity, transcriptional regulation, signal transduction, and subcellular localization in plants, mainly depending on the selective degradation of target proteins [8,31]. Zhu et al. identified a total of 1638 ubiquitination modification sites on 916 proteins in young rice spikes, which revealed that protein ubiquitination played a critical role in anther development and seed maturity [32]. A total of 234 differentially expressed proteins (DEPs) and 120 ubiquitinated DEPs were screened in peonies (*Paeonia suffruticosa*) under high temperature stress, and the ubiquitination of these proteins regulated hormone metabolism, flavonoid synthesis, and glycolysis [33]. After an 8 h drought treatment, more than 300 ubiquitination sites were found in potato (*Solanum tuberosum*) plants, which indicated that improved ubiquitination modifications could be an important adaptive response to drought stress [34].

UPS is the most dominant pathway for maintaining protein homeostasis related to seed germination [35], immune responses [10], and hormone signaling [36] in plants, since more than 80% of intracellular proteins are degraded via the UPS [1]. The UPS consists of six components including Ub, ubiquitinases (E1, E2, and E3), DUB, and 26S proteasome (Figure 2) [31,37]. Ub is a highly conserved globular protein with a molecular weight of 8.5 kDa, which was firstly discovered in 1975 [12,31]. Amino acid sequences of Ub are absolutely conserved in higher plants containing 76 amino acid residues with a signature terminal diglycine sequence (Gly75-Gly76). In addition, the number of amino acids of Ub is only two or three less or more among animals, plants, and fungi [26,27]. Seven conserved lysine (Lys) residues of Ub (Lys6, Lys11, Lys27, Lys29, Lys33, Lys48, and Lys63) can form an isopeptide which connects with the Gly76 of another Ub to further form a Ub chain (Figure 1c,d) [38,39]. The highly stable Ub, with the richness of internal hydrogen bonds, guides the degradation of proteins as a discernible tag in the UPS. An acyl-phosphoric anhydride bond is formed between adenosine monophosphate (AMP) and the free inactive Ub with the help of adenosine triphosphate (ATP), and then the Ub is activated as a result of the generation of a high-energy thioester bond between the carboxy-terminal glycine (Gly) of the Ub and E1 cysteine (Cys), thereby leading to the formation of the E1-Ub complex. The activated Ub is transferred to the Cys residue on the E2 by ester exchange reaction to form the E2-Ub complex. The E3 exhibits a critical role in transferring Ub to the corresponding target protein. Firstly, the target protein and E2-Ub complex are bound to the E3 to form the protein–E3-E2-Ub complex, and then the Ub is transferred from the E2-Ub to the target protein. An isopeptide bond is generated accordingly between the C-terminal Gly of the Ub and the amino group (usually a lysine residue) in the target protein. The Ub can be further linked to other Ubs or different lysines in the target protein, thereby forming multimono-ubiqumitination, a linear polyubiquitination chain, and branching polyubiquitination chains (Figure 3). The 26S proteasome recognizes Ubs and hydrolyzes the peptide bond to release Ubs (Figure 3). The recycling utilization of Ub provides a quick response to unfavorable environments after being detached from proteins by the DUBs [1,17,19,26,31].

## 3. Plant E3 Ubiquitin Ligase

In addition to their function of specific identification of a substrate domain [1], E3 ligases are known to regulate hormone signaling [40], plant development [41], and response to biotic and abiotic stresses [37,42]. The E3 ligases can be categorized into single- and multi-subunit groups [22]. The single-subunit E3 ligases are categorized according to the presence of four different structure domains: RING (Really Interesting New Gene), U-box, HECT (Homology to E6-associated Carboxy-Terminus), and RBR (Ring Between Ring) [43]. The RING contains a C6HC zinc finger acting as a metastable activator of E2 [44]. The RING can directly connect to E2-Ub complex and target proteins [42]. The monosubunit RING E3 ligases have large numbers of members in plants [45]. As the most widely reported E3 ligase, the RING has an important role in adaptative responses to salt, cold, drought, and heat in plants [27,46]. The U-box protein contains a structural domain of approximately 70 amino acids and is firstly identified in *Arabidopsis thaliana*. The main role of the U-box is to bind and stabilize E2-Ub through salt bridges and hydrogen bonds and then to facilitate Ub transfer to target proteins [44]. Many U-boxes are involved in the regulation of hormone signaling and stress response in plants [47,48]. The HECT has a HECT domain consisting of 350 amino acids at the C-terminus. Differing from other E3 ligases, the Ub is linked to the cysteine residues of HECT to form the Ub-HECT, and then the Ub is transferred to the target protein [22,27]. The HECT subfamily is small, and only seven members were identified in *A. thaliana*. Some HECTs regulate the development of plant trichomes [49]. RBR is composed of two RING structural domains and an IBR (In Between RING). The amino-terminal end of RBR interacts with E2-Ub [27]. It has been reported to play a role in regulating plant hormone levels, signal transduction, and response to adversity stresses [50,51].

The multisubunit E3 ligases are classified according to the structural composition of their different subunits. The APC/C (Anaphase Promoting Complex/Cyclosome), CBC VHL (Cullin-Elongin-BC-VHL), and CRLs (Cullin-RING Ligases) are three main types of multi-subunit E3 ligases [11,43]. APC/C consists of at least 11 core subunits exhibiting catalytic modules of the enzyme and the recognition and binding of substrates, and the APC/C is also a key regulator of cell division cycle [44,52]. The CBC VHL is mainly composed of VHL, RBX1, and CUL2. However, there is no in-depth study on roles of its components in plants [53]. Furthermore, the CRLs family is normally categorized into four subfamilies: SCF (SKP1-Cullin1-F-box), BTB (Bric-a-brac-Tram track-Broad), DDB (DNA damage-binding domain), and APC (an-aphase-promoting complex) [1,22,31]. As the largest and most well-researched subfamily, the SCF is made up of four components: skeleton protein CUL1 (Cullin1), core catalytic protein RBX1, F-box for specific identification of the target protein, and SKP1 (S-phase Kinase-associated Protein 1) for connecting CUL1 and F-box (Figure 2) [1,22]. SKP1 is a scaffolding protein which is rich in 160 amino acid residues. The F-box has a conserved segment of 40–50 amino acid residues at the N-terminal domain, binding to the SKP1 subunit to form the complex of SKP1-F-box. CUL1 is the major unit in the SCF complex, and its N-terminal region has a long rod-like structure consisting of 415 amino acids that binds to the complex of the SKP1-F-box. Differently, the C-terminal region of the CUL1 is a globular structure consisting of 360 amino acids which combine with RBX1. The E2 carrying the activated Ub can be further ligated to the RBX1 (Figure 3). Variable C-terminal domains of different F-box proteins let the SCF interact with various target proteins (Figure 3) [1,22,54].

In plants, the F-box are categorized into different family members based on the structure of the C-terminal domains, such as FBK (Kelch structure), FBL (LRR repeat-rich structural domain), FBA-D (F-box structure-associated domain and F-box domain), FBW (WD40 repeat structure), FBT (Tub structure), FBP (Phloem Protein 2 domain), and other members (Figure 2) [55]. The Kelch is a characteristic motif of the FBK consisting of 44–56 amino acid residues with eight highly conserved residues [56]. Multiple Kelch repeats can form foliaceous propeller structures that interact with other proteins [29,57]. One of the FBK proteins is also called the KFB, because its amino acid sequence contains a specific Kelch domain-containing F-Box. The FBLs, FBAs, and FBKs exhibit the largest numbers of members in the F-box family in plants [58,59]. A large number of FBK members have been identified in plants up to now, such as 69 *TaFBK* genes in wheat (*Triticum aestivum*) [60], 19 *PeKFB* in bamboo (*Phyllostachys edulis*) [61], 44 *StFBKs* in potato [62], and 31 *SmKFB* in Danshen (*Salvia miltiorrhiza*) [63]. Although a large number of *FBK* genes have been identified in plants, and some of them play critical roles in plant growth, organ development, secondary metabolism and stress response, the functional study of *FBK* genes in plants is still in the early stages.

## 4. Function of E3 Ligase *FBK* Gene in Plants

### 4.1. The Role of FBK in Plant Growth and Development

Many studies have indicated that the *FBK* plays an important role in regulating plant height, development of floral organs, seed germination and production, and leaf senescence in different plant species (Table 1). For example, the overexpression of *OsFBK1* in rice significantly increased the size of anther and stigma, the number of floral organs, and seed weight, but decreased pollen viability and the size of spikelet [23]. Moreover, the OsFBK1 interacted with OsATL53-OsCCR14 to regulate the lignification of rice roots and anthers [64]. Zegeye et al. found that the *OsFBK4* could positively regulate rice plant height by promoting the size of internodal cells [65], and the overexpression of *OsFBK12* led to delayed seed germination, a slowdown in leaf senescence, and the enlargement of seed [66]. It has been found that the OsLP is also an F-box protein containing Kelch repeat sequences in rice, and its mutant with increased inflorescence branches significantly increased seed yield. Most importantly, the mutant had a sturdier stalk, more vascular bundles, and more upright leaves, in favor of the cultivation and management of rice [67]. In chickpea (*Cicer arietinum*), Jia et al. identified an *FBK* gene *CarF-box1* which exhibited differential expression during seed development, germination, and floral development [68]. The FBK protein CTG10 in *A. thaliana* could promote seed germination and seedling growth by targeting phytochrome-interaction factor PIF1 [69]. The *AtKFB20* differentially expressed in the stems and leaves of *A. thaliana*, and the highest transcript level was detected in young nodes, which indicated that the gene may be involved in the mediation of plant growth and development [24].

It is noteworthy that FBK-regulated plant growth and development are closely connected with changes in endogenous phytohormones such as cytokinin (CTK), gibberellin (GA), and ethylene (ETH) [70,71,72]. Chen et al. found that rice OsFBK12 could induce the degradation of OsSAMS1, known as a key enzyme for the biosynthesis of ETH, to reduce ETH content, thereby leading to delayed seed germination and leaf senescence [66]. The OsFBK4 positively regulated plant height by affecting GA signaling-related and biosynthetic genes [65]. On the contrary, an F-box Kelch repeat protein PmFBK2 from *Persicaria minor* negatively regulated GA signaling by mediating the degradation of GA receptor GID1b, resulting in reduced seed germination, rosette diameter, root and hypocotyl length, and seed weight [73]. The TML gene encoding an F-Box protein with repeat Kelch structure in *Lotus japonicus* negatively regulated legume–rhizobium symbiosis by inhibiting CTK signaling [74]. The OsLP encodes a Kelch repeat-containing F-box protein in rice plants, and its mutation could significantly improve panicle architecture and grain yield. Further findings showed that the OsLP interacted with CTK oxidase/dehydrogenase OsCKX2 to regulate CTK level [67]. In addition, the Kelch-F-box protein SAGL1 inhibited the salicylic acid (SA) biosynthesis, thereby regulating plant growth and development [75]. The study of Li et al. also showed that a Kelch repeat F-box E3 ligase gene *AtARKP1* promoted the ABA signaling pathway, and the ABA also could induce the expression of *AtARKP1* in *A. thaliana* [76].

**Table 1 ijms-26-00821-t001:** The function of proteins encoded by *FBK* genes related to plant growth and development. The numbers in parentheses indicate references related to relevant findings. The “/” in the table indicates that relevant information is not mentioned in these literatures.

Plant Species	Name	Location	Target Protein	Function	Reference
*Oryza sativa*	OsFBK1	/	OsNAC1OsATL53-OsCCR14	The size of anther and stigma, the number of floral organ, seed weight, pollen viability, size of spikelet, lignification of rice anther and root	[23,64,77]
	OsFBK4	Nucleus, plasma membrane	/	GA signaling-related and biosynthetic genes, plant height	[65]
	OsFBK12	Nucleus	OsSAMS1	Leaf senescence, seed size and grain number, ETH content	[66]
	OsLP	Endoplasmic reticulum	SKP1, OsCKX	Panicle architecture, grain yield, CTK level	[67]
*Arabidopsis thaliana*	ARKP1	Nucleus	ASK1, ASK2	Abscisic acid signaling	[76]
	CTG10	Nucleus	PIF1	Seed germination	[69]
	KEB20	Cytoplasm	PAL	Plant growth	[24]
	SAGL1	/	SARD1	SA biosynthesis	[75]
*Cicer arietinum*	CarF-box1	Nucleus	/	Seed development and germination	[68]
*Lotus japonicus*	TML	Nucleus	/	CTK signaling, nodule number	[74]
*Persicaria minor*	PmFBK2	/	Skp1, PmGID1b	GA signaling, seed germination, rosette diameter, root and hypocotyl length, seed weight	[73]

### 4.2. The Role of FBK in Secondary Metabolism

Secondary metabolites such as terpenoids, steroids, and phenolic compounds are non-essential organic compounds produced by plants. These secondary compounds, relevant enzymes, proteins, and genes form a complex regulatory network for growth, development, and resistance to unfavorable external environments throughout the plant life cycle [78]. The biosynthesis and catabolism of secondary metabolites regulated by ubiquitination have been widely reported in plants. Increasing numbers of studies also prove that the *FBK* directly regulates certain secondary metabolite pathways [16]. As the first key enzyme in the phenylpropane pathway, phenylalanine ammonia-lyase (PAL) catalyzes the deamination reaction of phenylalanine (Phe), which is a precursor for the synthesis of flavonoids, lignans, and phenylpropanes [79]. Zhang et al. found that *AtKFB01*, *AtKFB20*, *AtKFB50*, and *AtKFB39* negatively regulated phenylpropanoid production via PAL ubiquitination and subsequent degradation in *A. thaliana*, and the down-regulation of these *FBKs* significantly improved phenylpropanoid biosynthesis [24,80]. Similar findings were demonstrated in the study of Kurepa et al. who found that the overexpression of *KFB20* encoding a Kelch repeat F-box protein in *A. thaliana* significantly reduced the accumulation of phenylpropanoid by promoting the proteolysis of PAL [81]. The Kelch repeat F-box protein SAGL1 negatively regulated the abundance of PAL1 enzyme in *A. thaliana*; therefore, a large amount of anthocyanin and lignin accumulated when the *SAGL1* was mutated [82]. The study of Wang et al. also demonstrated that the Kelch domain-containing F-box protein SnRK1 was a negative regulator of phenylalanine biosynthesis involved in the degradation of PAL. Accordingly, the accumulations of PAL protein, soluble phenolics, and lignin polymers could be significantly promoted by down-regulating *AtSnRK1* expression [83]. Yu et al. also reported that the SmKFB5 protein controlled the degradation of PAL, hence the biosynthesis of phenolic acid was negatively regulated by the SmKFB5 in Danshen [84].

Genetic evidence has revealed that expression levels of a group of *Kelch Domain F-Box* genes (*KFB1*, *KFB20*, *KFB39*, and *KFB50*) were significantly up-regulated in glucosinolate biosynthesis mutants, resulting in phenylpropanoid deficiency, suggesting a crosstalk between the glucosinolate and phenylpropanoid pathways in *A. thaliana* [85]. The overexpression of *CmKFB* in casaba muskmelons (*Cucumis melo*) enhanced the breakdown of naringenin chalcone, but significantly increased accumulations of coumarin and general phenylpropanoids [86]. Chalcone synthase (CHS) is a rate-limiting enzyme catalyzing the first step of flavonoid biosynthesis. The *KFB^CHS^* is negatively related to the production of flavonoids in *A. thaliana* because the KFB^CHS^ protein interacts with CHS to induce its ubiquitination and degradation. Disruption of *AtKFB^CHS^* expression resulted in high accumulations of CHS and flavonoids in *atkfb^chs^* mutant lines under ultraviolet irradiation [87]. Multiple *StKFB* genes differentially expressed in different colored potato tubers, indicating their potential role in the regulation of anthocyanin biosynthesis [62]. Overexpression of grape (*Vitis vinifera*) *VviKFB07* in tobacco (*Nicotiana benthamiana*) reduced the contents of flavonols and anthocyanins in corolla. Further findings demonstrated that the VviKFB07 improved the synthesis of stilbene by mediating the ubiquitination and degradation of VviCHS [88]. In addition, Yang et al. found that 19 *PeKFBs* differentially expressed in different tissues of moso bamboo (*Phyllostachys edulis*), of which PeKFB9 regulated lignin polymerization by interacting with lignin-degrading peroxidase PeSKP1-like-1 and PePRX72-1 [61]. It has also been reported that the degradation of cinnamoyl-CoA reductase was mediated positively by the OsFBK1 and 26S proteasome pathway in rice plants, leading to a decrease in lignin deposition in secondary cell walls of root and anther [77]. The *A. thaliana Kelch repeat F-box protein* (*SAGL1*) regulated the proteasome-dependent degradation of ECERIFERUM3 which is a critical enzyme for the biosynthesis of cuticular wax. Disruption of *SAGL1* significantly increased the accumulation of wax in stems, leaves, and roots [89]. Transgenic *A. thaliana* overexpressing the *SKIP11*, encoding a Kelch-repeat F-box protein, significantly reduced the production of green leaf volatiles, which are important regulators of plant–insect interaction due to the role of *SKIP11* in negatively regulating the hydroperoxide lyase pathway [90]. In summary, KFB proteins can interact with other key enzymes in secondary metabolite pathways to affect the accumulation of secondary metabolites. Furthermore, most of the FBK proteins negatively regulate secondary metabolism-related enzymes (Figure 4).

### 4.3. The Role of FBK in Stress Response

When plants are subjected to unfavorable environmental conditions, a systemic defense system is activated to maintain protein homeostasis [19]. For the maintenance of normal physiological homeostasis in plant cells, protein conformation, degradation, and recycling are altered by the UPS in favor of a rapid response to diverse environmental stresses [43]. More and more studies have proven that E3 ubiquitin ligases play an important role in tolerance to various biotic stresses (pathogens, bacteria, insect pests, weeds, etc.) and abiotic stresses (drought, high temperature, heavy metals, etc.) [31,43]. Multiple studies have shown that various biotic and abiotic stresses could induce changes in expression levels of different *FBKs* in different plant species (Table 2). Some *FBKs* regulate plant disease and insect resistance as well as the tolerance to abiotic stress through remodeling secondary metabolism.

#### 4.3.1. Biotic Stress

Plants have developed a two-layered defense strategy to protect themselves from pathogen attack. The first layer of defense is regulated by cell-surface-localized pattern recognition receptors (PRRs), and the second layer of defense is known as effector-triggered immunity (ETI), involved in many effector proteins and disease resistance (R) proteins. These PRRs, effector proteins, and R proteins are often modified by ubiquitination, phosphorylation, acetylation, etc. [12]. It has been reported that the rust fungus (*Puccinia recondita* f. sp. *tritici*) significantly altered transcript levels of *TaFBKs* in wheat plants [60,91]. The *FBK* gene *BIG24.1* could be significantly up-regulated by exogenous SA, methyl jasmonate (MeJA), ETH, and ABA in grape after being exposed to *Botrytis cinerea* infection [92]. The tolerance to powdery mildew (*Erysiphe necator* Schw.) could be significantly enhanced by the overexpression of the *VpEIFP1* encoding an F-box/Kelch-repeat protein, which induced thioredoxin proteolysis in wild Chinese *Vitis pseudoreticulata* [93]. At2g44130 containing the F-box/Kelch-repeat domain made *A. thaliana* plants more susceptible to root-knot nematodes *Meloidogyne incognita* by negatively regulating PAL activity [94]. The OsFBK16 interacted with OsPAL1, OsPAL5, and OsPAL6 to induce their degradation via the UPS pathway, and the *OsFBK16* knockout significantly enhanced the blast resistance of rice (*M. oryzae*) [21]. Thiel et al. found that the BvFBK protein from sugarbeet (*Beta vulgaris*) interacted with the pathogenicity factor P25, negatively regulating the resistance to beet necrotic yellow vein virus [95]. The study of Roshan et al. demonstrated that the interaction between SlKFB and AV2 protein from tomato (*Solanum lycopersicum*) leaf curl Palampur virus decreased the stabilization of the SlKFB, leading to an increase in PAL activity in virus-infected tobacco plants, which could indicate that the accumulation of PAL was related to virus infection and resistance [96].

#### 4.3.2. Abiotic Stress

It has been widely reported that E3 ligases, including RING type and U-Box type, act as core components of the UPS to meditate the tolerance to various abiotic stresses such as drought, salt stress, heat stress, cold stress, and heavy metal in plants [43]. However, the functional mechanism of the *FBK* family associated with plants’ adaptability to abiotic stress is poorly understood. An earlier study by Jia et al. found that the expression of the chickpea *FBK* gene *CarF-box1* was significantly up-regulated by drought, salt stress, and the application of MeJA; however, it was down-regulated under heat and cold stresses [68]. When potato plants responded to a mechanical wound, the inhibition of *miR2111* could increase the expression level of *IbFBK*, leading to improved ubiquitination and degradation of IbCNR8 [97]. In wheat plants, it was found that the *TaFBK* differentially expressed in different tissues and could be significantly up-regulated by exogenous SA and MeJA, but significantly down-regulated by exogenous NaCl and polyethylene glycol (PEG) stress after 12 h of treatments [91]. A large number of *TaFBKs* in wheat leaves differentially responded to heat stress, drought, and their combination; moreover, exogenous SA, ABA, NaCl stress, and PEG-induced drought stress significantly induced the expression of the *TaFBK19* at different time periods [60]. Further findings showed that the Kelch domain of TaFBK19 directly interacted with the PAL, indicating that the TaFBK19-regulated stress tolerance might depend on the phenylpropane pathway [60].

A recent study by Li et al. demonstrated that salt stress, heat stress, and ABA treatment induced the expression of *AtSDR* encoding an F-box protein, but drought stress significantly inhibited its expression. Furthermore, *AtSDR*-overexpressed plants had increased salt tolerance and susceptibility to drought [98]. The Kelch repeat F-box protein SAGL1 regulated the UPS-dependent degradation of ECERIFERUM3, which is a key enzyme for the biosynthesis of cuticular wax in *A. thaliana*. Disruption of the SAGL1 increased the accumulation of wax in leaves, contributing to enhanced drought tolerance [89]. All 19 *PeKFBs* containing stress-related cis-elements in their promoters differentially expressed in leaves of moso bamboo in response to drought and cold stress [61]. Similarly, the knockdown of *AtKFB01*, *AtKFB20*, or *AtKFB50* in *A. thaliana* significantly improved the tolerance to ultraviolet light stress, owing to a greater accumulation of polyphenols [80]. These studies highlight the function of the *FBK* family associated with the biosynthesis of secondary metabolites such as wax and polyphenols. However, most studies only focused on changes in the expression levels of *FBKs* in different plant species in response to different abiotic stresses. Therefore, the potential role and underlying function of different *FBKs* in stress tolerance still remain for further in-depth investigation in the future.

**Table 2 ijms-26-00821-t002:** Different stresses induce changes in the expression levels of *FBK* genes in different plant species. The “Positive” indicates that the gene plays a positive role in resisting adverse condition. The “Negative” means that the gene plays a negative role in resistance to the adverse situation. The numbers in parentheses indicate references related to relevant findings.

Stress Type	Stress Sub-Type	Species	Genes	Function	Citation
Biotic stress	*Puccinia recondita* f. sp. *tritici*	*Triticum aestivum*	*TaFBKs*	-	[60,91]
	*Botrytis cinerea*	*Vitis vinifera*	*BIG24.1*	-	[92]
	*Erysiphe necator* Schw.	*V. pseudoreticulata*	*VpEIFP1*	Positive	[93]
	*Meloidogyne incognita*	*Arabidopsis thaliana*	*At2g44130*	Negative	[94]
	*M. oryzae*	*Oryza sativa*	*OsFBK16*	Negative	[21]
	Beet necrotic yellow vein virus	*Beta vulgaris*	*BvFBK*	Negative	[95]
	Tomato leaf curl Palampur virus	*Nicotiana benthamiana*	*SlKFB*	-	[96]
Abiotic stress	Salinity	*Cicer arietinum*	*CarF-box1*	-	[68]
	Salinity	*T. aestivum*	*TaFBK*	-	[91]
	Salinity	*T. aestivum*	*TaFBK19*	-	[60]
	Salinity	*A. thaliana*	*AtSDR*	Positive	[98]
	Drought	*C. arietinum*	*CarF-box1*	-	[68]
	Drought	*T. aestivum*	*TaFBK19*	-	[60]
	Drought	*A. thaliana*	*AtSDR*	Negative	[98]
	Drought	*A. thaliana*	*SAGL1*	Negative	[89]
	Drought	*Phyllostachys edulis*	*PeKFBs*	-	[61]
	Low temperature	*C. arietinum*	*CarF-box1*	-	[68]
	Low temperature	*P. edulis*	*PeKFBs*	-	[61]
	High temperature	*A. thaliana*	*AtSDR*	-	[98]
	Mechanical wound	*Ipomoea batatas*	*IbFBK*	-	[97]
	Hormone treatment	*T. aestivum*	*TaFBK*	-	[91]
	Hormone treatment	*T. aestivum*	*TaFBK19*	-	[60]
	Hormone treatment	*A. thaliana*	*AtSDR*	-	[98]
	Ultraviolet light	*A. thaliana*	*AtKFB01, AtKFB20, AtKFB50*	Positive	[80]

## 5. Summary and Prospect

Ubiquitination is an important PTM of proteins involved in protein degradation, subcellular localization, DNA damage repair, transcription regulation, and signal transduction, thus affecting plant growth and development as well as the tolerance to various biotic and abiotic stresses. The FBK is an essential component of multi-subunit E3 ligase SCF, which plays a key role in the specific recognition of target proteins for ubiquitination-dependent protein degradation in the UPS. Many *FBK* genes have been identified in different plant species, and they regulate plant growth and development through affecting endogenous phytohormones. In addition, *FBK*-mediated plant tolerance to biotic and abiotic stresses is associated with changes in secondary metabolites such as phenylpropanoid, phenolic acid, flavonoid, lignin, wax, etc. Although E3 ligases have received increasing attention worldwide, limited information is available on the regulatory role and potential function of multiple *FBKs* related to stress tolerance in plant species, since it is still a huge challenge to generate highly purified multisubunit E3 ligases. Currently, a functional platform for reconstituting SCFs has been constructed in eukaryotes, laying the foundation for exploring the possible mechanism of SCF in plants [99]. A diverse range of FBK proteins interact with their target proteins depending on specific species, different developmental stages, and various environmental conditions.

*FBK* genes are involved in the regulation of plant life cycles and stress tolerance, but their specific mode of operation is not well known under stressful conditions. In addition, FBKs participate in plant metabolism and stress response by effectively regulating protein abundance and signal transduction via the modulation of the protein degradation rate, which is their uppermost physiological implication. However, many unresolved research questions remain to be answered in the future. For example, do different *FBK* genes act as positive or negative regulators during the ubiquitination process under normal conditions, biotic, or abiotic stresses? How do different FBKs recognize different target proteins for ubiquitination? Do specific FBKs interact with proteins exclusively or broadly? Does FBKs-regulated ubiquitination modification affect the activity of FBKs and the generation of other signals or not? Are FBK genes functionally redundant and do they directly regulate certain metabolic pathways in plants or not? Future research will focus on investigating the pivotal function of different FBKs which are located in different subcellular fractions. This will help to build an understanding of the potential roles of multi-subunit E3 ubiquitin ligases in mediation of plant growth, development, and stress defense in higher plants.

## Figures and Tables

**Figure 1 ijms-26-00821-f001:**
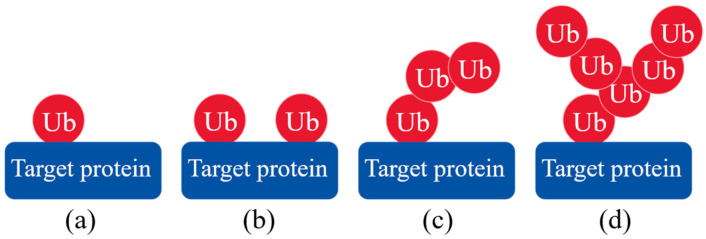
Classification of different types of ubiquitination processes: (**a**) mono-ubiquitination, (**b**) multimono-ubiquitination, (**c**) linear polyubiquitination, and (**d**) branching polyubiquitination.

**Figure 2 ijms-26-00821-f002:**
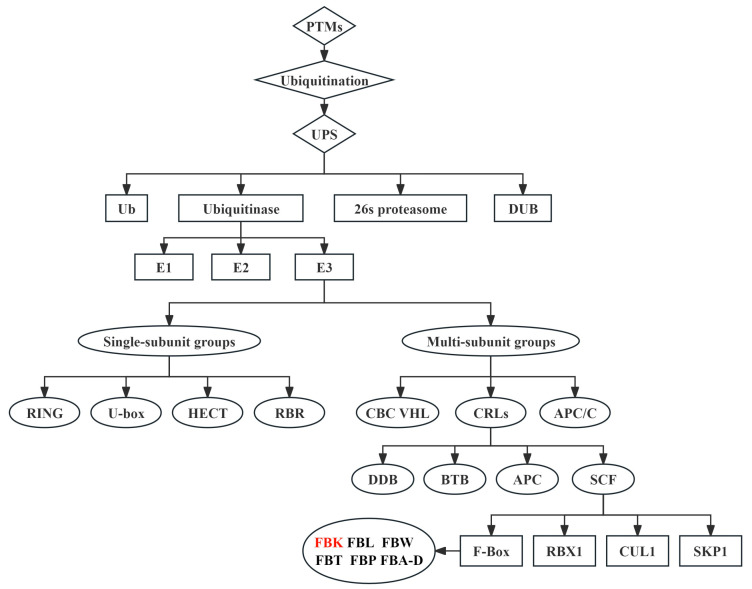
Post-translational modifications (PTMs), ubiquitination, and ubiquitin–26S proteasome system (UPS) in plants. Ub, ubiquitin; DUB, deubiquitinating enzyme; RING, Really Interesting New Gene; HECT, Homology to E6-associated Carboxy-Terminus; RBR, Ring Between Ring; CRLs, Cullin-RING Ligases; APC/C, Anaphase Promoting Complex/Cyclosome; CBC VHL, Cullin-Elongin-BC-VHL; SCF, SKP1-Cullin1-F-box; BTB, Bric-a-brac-Tram track-Broad; DDB, DNA damage-binding domain-containing; APC, an-aphase-promoting complex; CUL1, Cullin1; RBX1, RING Box-1; SKP1, S-phase Kinase-associated Protein 1; FBK, Kelch structure; FBL, LRR repeat-rich structural domain; FBW, WD40 repeat structure; FBT, Tub structure; FBP, Phloem Protein 2 domain; FBA-D, F-box structure-associated domain. A rectangular box represents a component which cooperates with other components to perform a function in the system, and an oval box represents a subfamily member which exhibits an independent function in the system. Text highlighted in red is the F-Box gene which is discussed in details in this review.

**Figure 3 ijms-26-00821-f003:**
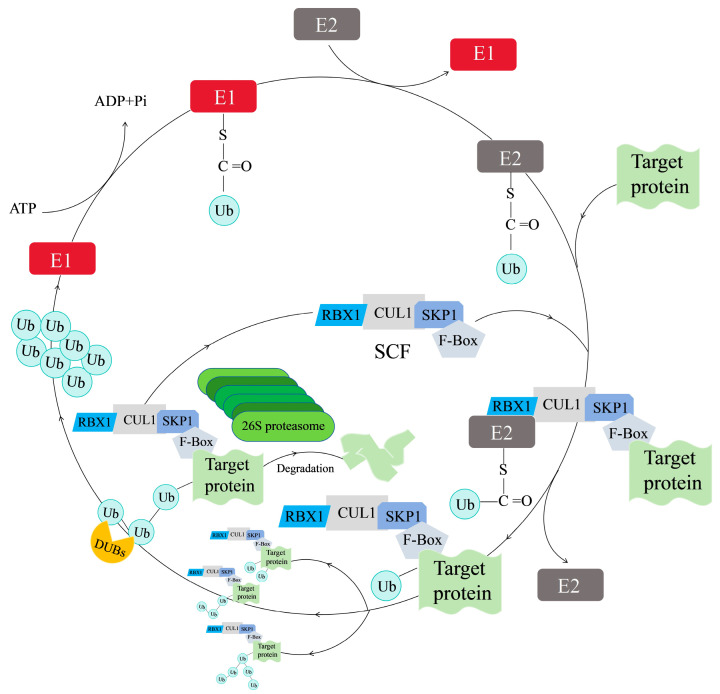
A working model of protein degradation depending on the ubiquitin–26S proteasome system (UPS) in plants. ADP, adenosine diphosphate; ATP, adenosine triphosphate; CUL1, Cullin1; DUB, deubiquitinating enzyme; E1, ubiquitin-activating enzyme; E2, ubiquitin-conjugating enzyme; E3, ubiquitin-ligating enzyme; RBX1, RING Box-1; SKP1, S-phase Kinase-associated Protein 1; SCF, Skp1-Cullin 1-F-box; Ub, ubiquitin.

**Figure 4 ijms-26-00821-f004:**
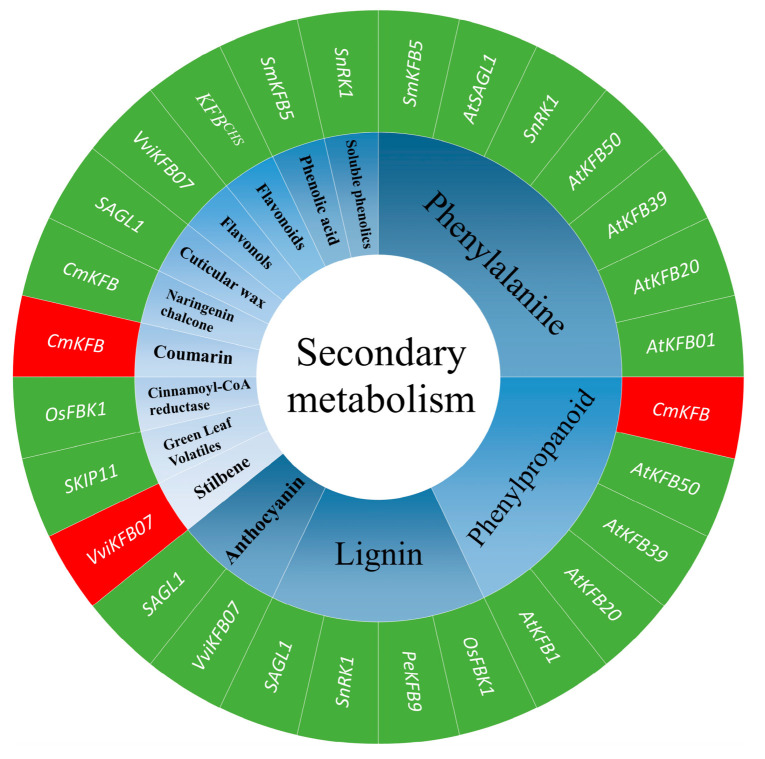
The function of *FBK* genes related to secondary metabolism in different plant species. The red or green background indicates that the gene positively or negatively regulates the biosynthesis of secondary metabolites, respectively: red=positive and green=negative. Different blue backgrounds indicate different secondary metabolites. All genes in the figure encode F-box protein with Kelch structures. The numbers in parentheses indicate references related to relevant findings. Genes and their correlative references: *AtSnRK1* [83]; *AtKFB01* [24,80,85]; *AtKFB20* [24,80,81,85]; *AtKFB50* [24,80,85]; *KFB39* [80,85]; *KFB^CHS^
* [87]; *CmKFB* [86]; *OsFBK1* [77]; *PeKFB9* [61]; *SAGL1* [82,89]; *SKIP11* [90]; *SmKFB5* [84]; *StFBK* [60]; *VviKFB07* [88].

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
