# Peer review of "The Role of E3 Ubiquitin Ligase Gene FBK in Ubiquitination Modification of Protein and Its Potential Function in Plant Growth, Development, Secondary Metabolism, and Stress Response"

_ijms, 2025, doi:10.3390/ijms26020821_

Round 1
Reviewer 1 Report
Comments and Suggestions for Authors
The authors mentioned the different sites for multi-ubiquitination, but the molecular-level details are missing. I suggest including a comprehensive figure of a ubiquitin structure labeled with possible linking sites and addressing different physiological consequences by each ubiquitination branch.
The authors show various types of E3 ligases in Figure 1 and address them at the beginning of section 3, but there is no detailed explanation of the characteristics of each class. For example, they could write briefly about the main differences between the HECK, RING, U-box, and RBR type E3 ligases, and the single-subunit and multi-subunit groups by focusing on what different physiological events is associated with each group.
In Figure 2, given the quality of images in the manuscript, the authors should consider replacing this figure in vector format before the publication.
In Figure 3, it can be assumed that the numbers in parentheses indicate the references related to each finding, but the authors should consider writing what those numbers mean in the figure legend.
In section 5, the authors should consider writing more about why study FBK E3 ligase in plants by addressing what has not been known and what is the physiological implication. The current version does not show many good reasons to study this particular protein species other than it is less known and was difficult to reconstitute in vitro.
Author Response
Comments and Suggestions for Authors
- The authors mentioned the different sites for multi-ubiquitination, but the molecular-level details are missing. I suggest including a comprehensive figure of a ubiquitin structure labeled with possible linking sites and addressing different physiological consequences by each ubiquitination branch.
Response: Thank you very much for professional review and giving us some good suggestions to improve our study. A comprehensive figure of a ubiquitin structure labeled with possible linking sites has been made and added as the new Figure 1 in the revised manuscript (Figure 1a-d). We have further explained mono-ubiquitination, multimono-ubiquitination, linear polyubiquitination, and branching polyubiquitination in revised manuscript according your suggestion (line 78-90).
- The authors show various types of E3 ligases in Figure 1 and address them at the beginning of section 3, but there is no detailed explanation of the characteristics of each class. For example, they could write briefly about the main differences between the HECK, RING, U-box, and RBR type E3 ligases, and the single-subunit and multi-subunit groups by focusing on what different physiological events is associated with each group.
Response: Thank you very much for good suggestions. More explanation of characteristics of each class and their main differences have been added in details in the revised manuscript (line 151-178).
- In Figure 2, given the quality of images in the manuscript, the authors should consider replacing this figure in vector format before the publication.
Response: Thanks. High-resolution images have been added in revised manuscript.
- In Figure 3, it can be assumed that the numbers in parentheses indicate the references related to each finding, but the authors should consider writing what those numbers mean in the figure legend.
Response: Thank you very much for good suggestion. The meaning of these numbers in parentheses has been added in legends of Table 1, Table 2, and Figure 3: The numbers in parentheses indicate references related to relevant findings.
- In section 5, the authors should consider writing more about why study FBK E3 ligase in plants by addressing what has not been known and what is the physiological implication. The current version does not show many good reasons to study this particular protein species other than it is less known and was difficult to reconstitute in vitro.
Response: Thanks. In section 5, we have added reasons for studying FBK ligase and its research implications. Its significance is broadened by the addition of currently unresolved research questions (line 397-431).
Reviewer 2 Report
Comments and Suggestions for Authors
This manuscript provides the critical role of protein ubiquitination, in regulating plant growth, development, and responses to environmental stresses. Specifically, it offers a detailed analysis of the ubiquitin-proteasome system (UPS), with a focus on the FBK gene's unique contributions within the ubiquitin ligase system. Overall, the manuscript is well-structured and comprehensive, presenting valuable insights into the mechanism of FBK-mediated ubiquitination and its potential applications in plant biology. It serves as an informative resource for researchers investigating the interplay between protein regulation and plant physiology.
Comments and Suggestions for Improvement:
Lines 57–67: A progressive narrative structure is recommended. These lines should be integrated with Lines 28–35 for a cohesive discussion.
Line 73: Clearly distinguish monoubiquitination, multiubiquitination and polyubiquitination. Discuss their differences and implications in detail to enhance clarity.
Line 94: In fact, not all species of ubiquitin have 76 amino acids, so it's important to be clear here.
Lines 135–136: Clarify the origin of abbreviations such as FBD and FBA. Ensure their derivation aligns with their full names for consistency and precision.
Lines 218–219: Address the naming ambiguity between AtKFB and FBK. If they refer to the same entity, clarify this earlier in the manuscript to avoid confusion.
Figure 3: Verify whether SnRK1 and SAGL1 are members of the FBK family. Provide appropriate annotations or clarifications for the figure.
Line 344: The summary and outlook section lacks depth and primarily repeats prior content. Restructure this section to offer layered insights, addressing future research directions and broader implications systematically.
By addressing these points, the manuscript can achieve greater clarity, depth, and impact, ensuring its significance within the field of plant biology is fully realized.
Author Response
Comments and Suggestions for Authors
This manuscript provides the critical role of protein ubiquitination, in regulating plant growth, development, and responses to environmental stresses. Specifically, it offers a detailed analysis of the ubiquitin-proteasome system (UPS), with a focus on the FBK gene's unique contributions within the ubiquitin ligase system. Overall, the manuscript is well-structured and comprehensive, presenting valuable insights into the mechanism of FBK-mediated ubiquitination and its potential applications in plant biology. It serves as an informative resource for researchers investigating the interplay between protein regulation and plant physiology.
Response: Thank you very much for professional review and giving us some good suggestions to improve our study. We have revised current manuscript according all suggestions.
Comments and Suggestions for Improvement:
- Lines 57–67: A progressive narrative structure is recommended. These lines should be integrated with Lines 28–35 for a cohesive discussion.
Response: Thank you very much for good suggestion. We have integrated these two parts in revised manuscript according to your suggestion (line 29-42).
- Line 73: Clearly distinguish monoubiquitination, multiubiquitination and polyubiquitination. Discuss their differences and implications in detail to enhance clarity.
Response: Thank you very much for good suggestion. We have further explained monoubiquitination, multiubiquitination, and polyubiquitination and also discussed their differences and implications in details in revised manuscript (line 78-90; Fig 1a-d). In addition, we also proved a new Figure 1 to distinguish them.
- Line 94: In fact, not all species of ubiquitin have 76 amino acids, so it's important to be clear here.
Response: Thanks. Up to now, almost all researches show that the amino acid sequence of ubiquitin is absolutely conserved in higher plants, and a ubiquitin includes 76 amino acids. However, the number of amino acids of a ubiquitin is only two or three less or more among animals, plants, and fungi. We also added relevant explanation in revised manuscript (line 113-119).
- Lines 135–136: Clarify the origin of abbreviations such as FBD and FBA. Ensure their derivation aligns with their full names for consistency and precision.
Response: Thank you very much for careful review. The FBA and FBD are the same E3 ligase with an F-box characteristic structural domain. So, it often is identified as as FBA-D, and we have modified it in the revised manuscript (line 196-197).
- Lines 218–219: Address the naming ambiguity between AtKFB and FBK. If they refer to the same entity, clarify this earlier in the manuscript to avoid confusion.
Response: Thank you very much for good suggestion. Yes, both FBK and KFB denote specific proteins with Kelch structure and F-box. One of FBK proteins is also called the KFB, because its amino acid sequence contains specific kelch domain-containing F-Box. Relevant explanation has been added in revised manuscript (line 201-202).
- Figure 3: Verify whether SnRK1 and SAGL1 are members of the FBK family. Provide appropriate annotations or clarifications for the figure.
Response: Thank you very much for professional and careful review. Yes, both SnRK1 and SAGL1 genes are members of the FBK family, and they were only renamed by authors. We have provided appropriate annotations or clarifications for the figure. Please see the legend of Figure 4 (line 320-321).
- Line 344: The summary and outlook section lacks depth and primarily repeats prior content. Restructure this section to offer layered insights, addressing future research directions and broader implications systematically.
Response: Thank you very much for professional review and good suggestion. We have revised the summary and outlook section to address future research directions and broader implications systematically (line 397-431).
- By addressing these points, the manuscript can achieve greater clarity, depth, and impact, ensuring its significance within the field of plant biology is fully realized.
Response: Thank you very much for professional and careful review again. Your suggestions help us to improve this review.
Round 2
Reviewer 2 Report
Comments and Suggestions for Authors
The revised manuscript has significantly improved in quality; however, the following issues need to be addressed for further refinement:
1. Figure 2:
(1)The layout requires modification for better clarity and readability. For instance, the subsequent processes or fuction of multimono-ubiquitination and branching polyubiquitination need further elaboration.
(2)The meaning of the red arrow on the left should be clarified or explained in the figure legend.
(3)In the lower left corner, the label "RBX1" is obscured by overlapping arrows, which may lead to confusion and should be corrected.
2. Table 1:
Gene names and species names should not break across lines to maintain readability and ensure a clean presentation.
3. References:
The format of the references should be carefully checked for consistency. Specific issues include:
Reference 4: The species name Arabidopsis should be italicized.
Reference 45: There is inconsistency in font styles.
Reference 66: Gene names should be italicized.
Addressing these points will ensure improved presentation and adherence to publication standards.
Author Response
Comments and Suggestions for Authors
The revised manuscript has significantly improved in quality; however, the following issues need to be addressed for further refinement:
Response: Thank you very much for professional review. These mistakes have been corrected according your suggestions.
- Figure 2(new Figure 3):
(1)The layout requires modification for better clarity and readability. For instance, the subsequent processes or function of multimono-ubiquitination and branching polyubiquitination need further elaboration.
Response: Thank you very much for careful review. The layout has been improved for better clarity and readability in new Figure 3. In addition, we have added the subsequent processes of multimono-ubiquitination, linear polyubiquitination, and branching polyubiquitination to elaborate the Figure 3 in revised manuscript (line 69-81; 121-126).
(2)The meaning of the red arrow on the left should be clarified or explained in the figure legend.
Response: Thanks for careful review. The red arrow has been deleted.
(3)In the lower left corner, the label "RBX1" is obscured by overlapping arrows, which may lead to confusion and should be corrected.
Response: Thanks. we have changed ambiguous parts and removed those overlapping.
- Table 1:
Gene names and species names should not break across lines to maintain readability and ensure a clean presentation.
Response: Based on your suggestions, we have adjusted the format of the species and protein names in the table 1. To present the table 1 and 2 clearly, we have also put a blank line between the different species.
- References:
The format of the references should be carefully checked for consistency. Specific issues include:
Reference 4: The species name Arabidopsis should be italicized.
Reference 45: There is inconsistency in font styles.
Reference 66: Gene names should be italicized.
Response: Thank you very much for your careful review. We have revised all the reference format.
Addressing these points will ensure improved presentation and adherence to publication standards.
Response: Thank you very much for professional review. We sincerely appreciate your time and hard work.